# An Improved Algorithm for Extracting Subtle Features of Radiation Source Individual Signals

**Jingchao Li [1], Dongyuan Bi [1], Yulong Ying [2],\*, Kai Wei [1] and Bin Zhang [3]**

[1]   School of Electronic and Information, Shanghai Dianji University, Shanghai 201306, China;
     lijc@sdju.edu.cn (J.L.); bdy_521@163.com (D.B.); wk00528@126.com (K.W.)
[2]   School of Energy and Mechanical Engineering, Shanghai University of Electric Power, Shanghai 201306,
     China
[3]   Department of Mechanical Engineering, Kanagawa University, Yokohama 221-8686, Japan;
     zhangbin@kanagawa-u.ac.jp
\*    Correspondence: yingyulong060313@163.com

**Abstract:** With the rapid development of communication and information technology, it is difficult for traditional signal detection and recognition methods to accurately acquire and identify the intelligence under complex environments. In order to solve this problem, this paper proposes a subtle feature extraction and recognition algorithm for radiation source individual signals based on multidimensional hybrid features. Firstly, Hilbert transform was performed on the radiation source signals from 10 identical radio devices, and the subtle features of different radiation sources' signals were extracted. Then, traditional principal component analysis (PCA) algorithm was used to extract and reduce the principal components of the extracted feature data sets. Aiming at the insufficiency of traditional PCA algorithm, an improved principal component analysis algorithm was proposed. At last, a gray relation algorithm was used to classify and identify the radiation source individual signals, and the recognition rate was calculated. Experimental results show that Hilbert transform combined with the improved PCA algorithm can achieve a recognition rate of 99.67% for the "fingerprint" features of radiation source individual signals under the signal-to-noise ratio (SNR) of 20 dB. Compared with the traditional algorithms, the recognition rate increased by 5.67%. Therefore, it provides a powerful theoretical basis for extracting subtle features of radiation source devices under complex electromagnetic environments.

**Keywords:** subtle feature extraction; signal recognition; Hilbert transform; improved PCA algorithm; gray relation classifier

## 1. Introduction

With the rapid development of communication and information technology, the limitations of network security control are increasing, and researchers have a deeper understanding of the security and supervisability of information in wireless networks. In military, it is difficult for traditional methods to accurately acquire and identify intelligence in the existing complex environments. Therefore, research on the extraction of radiation source's subtle features and recognition technology attracts more researchers' attention. Among them, the extraction technology for radiation source individual signals is a key technology in the field of non-cooperative communication. How to select a feasible and efficient signal processing method to accurately analyze and extract subtle features is the hot spot and difficulty of this technology.

As a new science and technology, individual radiation source identification technology has attracted great attention from researchers in various countries. Starting from the generation mechanism of radiation

source individual characteristics, the research ideas of individual radiation source identification mainly focuses on the analysis of the transient characteristics and steady-state characteristics of signals [1,2]. Initially, individual radiation source identification was mostly directed to radar emitter signals [3–5]. In recent years, with the continuous improvement of the technology level of electronic devices, the subtle differences between same type of radio stations are gradually reduced. In the harsh communication environment, the subtle features of signals are easily obliterated by noise, and the detection and extraction of their features have become a problem which makes people pay more attention. The traditional radiation source individual recognition algorithms have been widely used. Based on the characteristic parameters of high-order cumulant [6–8], the translation, scale and phase rotation of the signal constellation are uncertain, which has been well applied in individual signal recognition. But when the data length is limited and the noise is not steady, it is difficult to distinguish the individual signals that are to be identified. The characteristic parameters of the signal power spectrum [9,10] and the high-power spectrum [11,12], have better anti-noise performance than time-domain statistical parameters [13–15]. However, the nonlinear transformation of signals will destroy the validity of the features and reduce the separability of the extracted feature. The Principal Component Analysis (PCA) algorithm is often applied to face recognition [16,17] and chemical process monitoring [18], with good recognition effects. The traditional PCA algorithm is mainly applied to regression optimization problems, such as data processing and dimensionality reduction [19].

According to the relevant theories and actual needs, this paper proposes a PCA feature extraction algorithm based on Hilbert transform and its improved algorithm. The feature extraction of the radiation source individual signals was carried out, and then the gray relation theory was used to recognize the extracted features. The extracted features were classified to achieve the purpose of identifying the "fingerprint" features of the radiation source individual signals.

## 2. Methodology

With the rapid development of communication information technology and Internet industry, the technology for the extraction and recognition of radiation source individual subtle features has a wide application prospect in military electronic warfare field and civil fields.

This paper proposes a principal component analysis (PCA) feature extraction algorithm and its improved algorithm, which are based on Hilbert transform. A gray relation classifier is used to classify the extracted features. The radiation source individual subtle feature extraction and recognition schematic diagram is shown in Figure 1. The practical transient signals from 10 identical radio devices from the same manufacturer are taken as example. Firstly, the radiation source transient measured signals are sampled. Then, the Hilbert transform is performed on the sampled emitter signals. At the same time, the principal component analysis algorithm and its improved algorithm are used to extract the robust subtle features of the radiation source transient measured signals. Finally, the gray relation algorithm is used to classify the emitter signals, and the recognition rate is calculated. The effectiveness of the proposed algorithm is verified by comparing the recognition results with traditional algorithms.

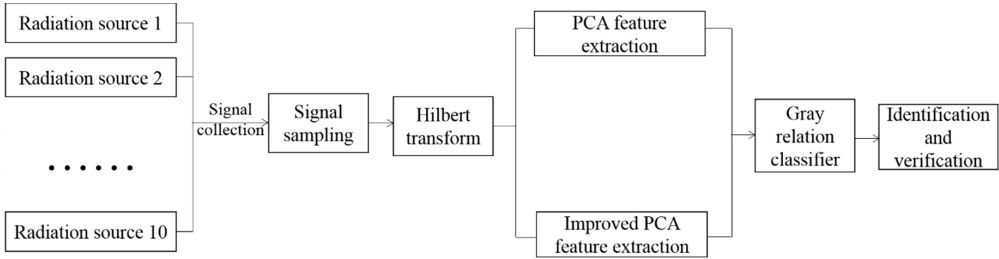

**Figure 1.** Schematic diagram of the extraction and recognition of radiation source individual subtle features.

### 2.1. Principal Component Analysis Theory

Principal Component Analysis (PCA) is a statistical analysis method that uses the idea of dimension reduction to transform original variables into a few new comprehensive variables through linear

transformation [20]. These new variables (principal components) are not related to each other and can effectively represent the information of the original variables. The core problem of principal component analysis is how to find the maximum variance of derived variables and the linear combination of the original data set by PCA algorithm. In the study, the method used in the paper is to obtain the singular value decomposition (SVD) of the data matrix, and then calculate the PCA by SVD.

Suppose that $X$ is an $n \times p$ dimensional matrix. Where $n$ is the number of samples and $p$ is the number of variables. For the generality of the calculation, let the mean value of each column of the matrix be equal to 0. According to the formula, the singular value decomposition of $X$ matrix can be expressed by the following equation [21].

$$X = UDV^T \tag{1}$$

The matrix $D$ is a rectangular diagonal matrix with size $n \times p$, which is an $X$ singular value. The matrix $U$ is an $n \times n$ matrix, which is a left singular matrix of $X$. The column of matrix $U$ is an orthogonal unit vector with length $n$. The matrix $V$ is a $p \times p$ matrix, which is a right singular matrix of $X$, and the column of matrix $V$ is an orthogonal unit vector. Then, according to the singular value decomposition theory, a partition matrix could be obtained.

$$Y^T = X^T U = VD^T U^T U = VD^T \tag{2}$$

where the left singular vector of $X^T$ is $V$. The polar decomposition of $Y^T$ is also expressed in this way.

In practical algorithm, it is often not necessary to form a matrix $X^T X$ to calculate the SVD of $X^T$. Therefore calculating principal component from data matrix is the standard method to calculate SVD unless we need a small amount of components.

Then the principal component analysis algorithm transforms the original data set into a new coordinate system and produces the largest variance. Where the projection of the data set on the first coordinate is called the first principal component. Similarly, projection on the second coordinate is called the second principal component.

Finally, by multiplying the first one singular value with its corresponding singular vector, the score matrix $Y_l$ can be obtained, which is reduced to an $n \times l$ dimensional matrix.

$$Y_l = U_l^T X = D_l V^T \tag{3}$$

By using the principal component analysis algorithm, the original data set is transformed into an $n \times l$ dimensional matrix.

### 2.2. Gray Relation Algorithm

The principle of gray relation analysis (GRA) [22,23] is to compare and analyze the similarity of geometric shapes between different curves. According to the similarity, the relation coefficient and the relation degree between the system variables are calculated. The smaller the geometric difference between the input sequence curves, the greater the relation degree.

Assume that the behavior sequence of a system is as follows:

$$
\begin{aligned}
X_0 &= (x_0(1), x_0(2), \cdots, x_0(n)) \\
X_1 &= (x_1(1), x_1(2), \cdots, x_1(n)) \\
&\quad \cdots \\
X_i &= (x_i(1), x_i(2), \cdots, x_i(n)) \\
&\quad \cdots \\
X_m &= (x_m(1), x_m(2), \cdots, x_m(n))
\end{aligned}
\tag{4}
$$

$X_0$ is the input sequence to be compared; $X_1$, $X_2$, $X_m$ are priori reference sequences. The variable $i = 0, 1, 2, \cdots, m$ is used to represent the type of known communication signals in the database, and $n$ is the number of the signal's characteristic parameters.

By definition:

$$r(x_0(j), x_i(j)) = \frac{min_i min_j |x_0(j) - x_i(j)| + \rho max_i max_j |x_0(j) - x_i(j)|}{|x_0(j) - x_i(j)| + \rho max_i max_j |x_0(j) - x_i(j)|} \tag{5}$$

$$\gamma_{0i} = \frac{1}{n} \sum_{j=1}^{n} \gamma(x_0(j), x_i(j)) \tag{6}$$

$$i = 1, 2 \cdots, m, \ j = 1, 2 \cdots, n \tag{7}$$

In the upper formula, $\rho$ is the distinguishing coefficient, and this variable is usually set to 0.5 in the actual algorithm; $\gamma(x_0, x_1)$ is the gray relation degree between $X_0$ and $X_i$, which is usually expressed by the letter $\gamma_{0i}$. In addition, $\gamma(x_0(j), x_i(j))$ is the $j$th gray relation coefficient which is often expressed by $\gamma_{0i}(j)$.

Then the complete gray relation algorithm is introduced. The specific steps are as follows:

(1) obtain an image of the initial value of each variable $x_i'$ to be tested, with the following formula:

$$x_i' = \frac{x_i}{x_i(1)} = (x_i'(1), x_i'(2), \cdots, x_i'(n)), i = 0, 1, 2, \cdots, m \tag{8}$$

where $i = 0, 1, 2, \cdots, m$ represents the type of known signal in the database; $n$ is the number of signal features.

(2) obtain the sequence of differences as follows:

$$\Delta x_i(k) = x_0'(k) - x_i'(k), \Delta x_i = (\Delta x_i(1), \Delta x_i(2), \cdots, \Delta x_i(n)), i = 1, 2, \cdots, n \tag{9}$$

where $j = 0, 1, 2, \cdots, n$ represents the $j$th signal characteristic parameter.

(3) Find the maximum difference M and the minimum difference m, that is:

$$M = max_i max_j \Delta x_i(j) \tag{10}$$

$$m = min_i min_j \Delta x_i(j) \tag{11}$$

(4) Calculate the relation coefficient value $\gamma_{0i}(j)$, that is:

$$\gamma_{0i}(j) = \frac{m + \rho M}{\Delta_i(j) + \rho M} \quad \rho \in (0, 1) \tag{12}$$

$$j = 1, 2, \cdots, n \ \ i = 1, 2, \cdots, m$$

(5) Calculate the relation degree between sequences $\gamma_{0i}$, that is:

$$\gamma_{0i} = \frac{1}{n} \sum_{j=1}^{n} \gamma(x_0(j), x_i(j)) \tag{13}$$

where, $\gamma_{0i}$ represents the similarity degree between different sequences.

## 3. Improve PCA Algorithm

When classifying a radiation source individual signals, it is often necessary to remove the mean value. The reason is that sometimes the data set is far from the average value, which causes a large error in the results of the principal component analysis. In this paper, an improved algorithm is

proposed for the traditional PCA method. Another mathematical algorithm of mean value is used to preprocess the data set of the communication signal to improve the validity of the algorithm.

In theory, any data set consists of two parts of information, among them: (1) the difference information that reflects the degree of dispersion of each indicator (reflected by the variance of each indicator); (2) relevant information on the degree of interaction among the indicators (reflected by the correlation matrix). In the traditional PCA algorithm, the standardized method is to make the variance of each indicator be equal to 1. That is to say, the diagonal elements in the covariance matrix of the matrix are all equal to 1. The difference in the degree of variation of the original data can be eliminated. But the drawback of this approach is also obvious: after the raw data is standardized, in subsequent principal component extraction, the extracted principal components only contain the information of the interaction of the data. However, because of the lack of different information on the degree of variation of each indicator, this standardized processing method cannot completely reflect all the information of the original signals. In order to solve this problem, this paper proposed an improved method based on the meanization of data.

First of all, the original indicator data is transformed into the same trend, and the results are obtained: $X = (x_{ij})_{n \times p}$, and then:

$$Z_{ij} = \frac{x_{ij}}{\overline{x_j}}, \ i = 1, 2, \cdots, n; \ j = 1, 2, \cdots, p; \tag{14}$$

where $\overline{x_j} = \frac{1}{n} \sum_{l=1}^{n} x_{ij}$. At this point, the meanization matrix $Z = (z_{ij})_{n \times p}$ can be calculated; And then, the sample covariance matrix $S = (S_{ij})_{p \times p}$ of the meanization matrix Z can be calculated.

$$s_{ij} = \frac{1}{n-1} \sum_{l=1}^{n} (z_{li} - z_i)(z_{lj} - z_j) \tag{15}$$

where $z_i = \frac{1}{n} \sum_{l=1}^{n} (z_{li})$. Finally, the principal component analysis is carried out using covariance matrix as sample.

Through the analysis of the above steps, the improved principal component analysis algorithm can not only completely represent most of the effective information of the original data set, but also effectively avoid the problem of missing information in the traditional PCA algorithm. In addition, reference [24] shows that the relation coefficient between indicators is not changed by the meanization treatment, and all the information of the relation coefficient matrix is reflected in the corresponding covariance matrix. The covariance matrix after meanization processing not only eliminates the influences of the dimension and magnitude of the index,, but also contains all the information of the original data. Therefore, when the principal component analysis method is used for comprehensive evaluation, the meanization method is applied to dimensionless processing. Therefore, the principal component analysis (PCA) algorithm with the meanization method is helpful to eliminate the outliers, but it does not change the superiority of the relation coefficient between indicators. That is to say, all the information of the relation coefficient matrix is reflected in the corresponding covariance matrix.

## 4. Experimental Results and Analysis

In this paper, 500 sets of transient signals from 10 identical Motorola walkie-talkies from the same manufacturer were collected. Data were collected with Agilent oscilloscope and were injected directly without antenna. Fifty sets of data were collected for each radio device at a sampling rate of 40 MHz, and each set of data contained 159,901 data points. Randomly selected signal waveforms of four groups of radio devices are shown in Figure 2a–d.

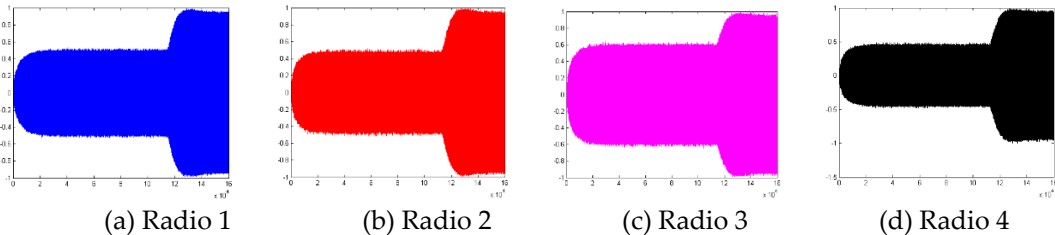

(a) Radio 1　　　　　　(b) Radio 2　　　　　　(c) Radio 3　　　　　　(d) Radio 4

**Figure 2.** Randomly selected signal waveforms of four groups of radio devices.

It can be seen from the experimental results shown in Figure 2 that the RF signal time-domain waveforms of these 10 identical radio devices seem almost the same. After extracting 10 sets of subtle features of the source' and select their two-dimensional principal components using the Hilbert transform and the PCA algorithm in a noise-free communication environment. The related feature distribution diagram is shown in Figure 3.

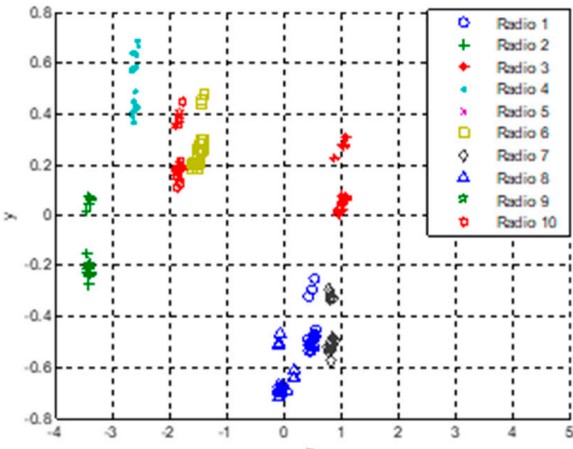

**Figure 3.** Two-dimensional subtle features of the principal components of 500 sets of data from ten identical radio devices based on Hilbert transform and the PCA algorithm in a noiseless communication environment.

It can be seen from the experimental results in Figure 3 that, on the basis of the Hilbert transform and the PCA algorithm in a noiseless communication environment, the subtle features from the time–domain waveforms can be effectively extracted. The different signals in the figure have good inter-class separation and intra-class aggregation. The *X* axis is the first principal component and the *Y* axis is the second principal component value. Through data analysis, the contribution rate of the first principal component is 95.83%, and the contribution rate of the second principal component is 2.76%. The cumulative contribution rate of these two principal components is 98.59%. According to the gray relation algorithm, the recognition rate of these 500 sets of transient signals from 10 identical Motorola walkie-talkies is calculated to be 100%. That is to say the proposed method can realize accurate classification of similar signals from different emitter sources.

The noiseless communication environment is further changed to a noisy communication environment with the SNR of 20 dB. The experimental results are shown in Figure 4 according to the above operation.

From the experimental results, it can be seen that the subtle features of the identical radio devices can be effectively extracted on the basis of the proposed Hilbert and PCA method in a noisy communication environment with the SNR of 20 dB.

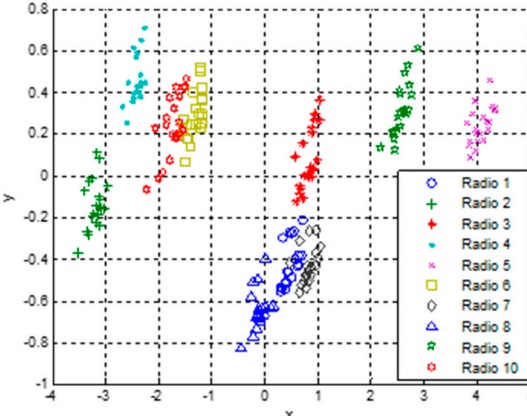

**Figure 4.** Two-dimensional subtle features of the principal components of 500 sets of transient data from ten identical radio devices based on Hilbert transform and the PCA algorithm in a noisy communication environment with the SNR of 20 dB.

As shown in Figure 4, in a noisy communication environment with the SNR of 20 dB, a few part of the subtle features overlap. That is to say, it is impossible to achieve accurate classification and recognition with recognition rate of 100%. Through increasing the number of principal components, the recognition rate can be increased. When the dimension of the features increases, the cumulative principal component contribution rate will increase and the recognition rate will be improved. In practical applications, researchers can choose appropriate dimensions of the principal components according to their requirements. Through data analysis, the contribution rate of the first principal component is 76.64%, and the contribution rate of the second principal component is 2.28%, and the contribution rate of the third principal component is 0.66%, and the contribution rate of the fourth principal component is 0.4%. The experimental results show that the cumulative contribution rate of the principal components is only 78.92%, and the recognition rate is 94%. In order to improve the effectiveness of the proposed method, two other principal components are added. That is to say, four principal components are used together as the characteristic parameters of 500 sets of transient data. The cumulative contribution rate of the four principal components is 79.98%, and the recognition rate is increased to 99.67%.

In order to solve the problem of low contribution rate of two-dimensional principal components in in a noisy communication environment with the SNR of 20 dB, an improved PCA algorithm is proposed. The improved PCA algorithm with meanization is applied to the recognition system and the results are shown in Figure 5.

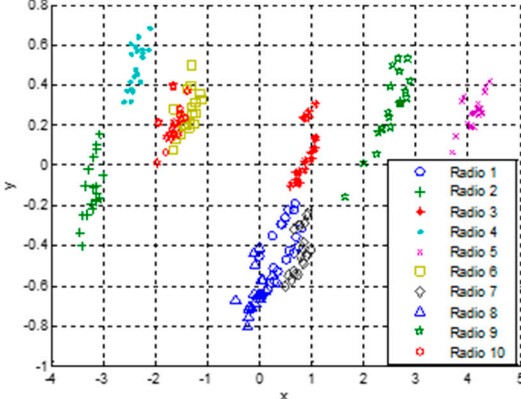

**Figure 5.** Two-dimensional subtle features of the principal components of 500 sets of transient data from 10 identical radio devices based on Hilbert transform and the improved PCA algorithm in a noisy communication environment with the SNR of 20 dB.

The simulation results show that the recognition rate based on the Hilbert transform and the improved PCA algorithm is 99.67% under a noisy communication environment with the SNR of 20 dB. In the figure, the first principal component is represented by the $X$ axis of the abscissa coordinate and the second principal component is represented by the $Y$ axis of the vertical coordinate. Compared with the traditional algorithm, the recognition rate is increased by 5.67%.

According to the experimental results above, the performance of Hilbert transform with traditional PCA and the improved PCA algorithm are summarized as follows:

In a noise-free environment, the recognition rate based on Hilbert transform and traditional PCA algorithm can reach 100% and the contribution rate of the first principal component has exceeded 95%. In this case, the advantages of the improved PCA algorithm based on Hilbert transform are not obvious. However, in the noisy environment with the SNR of 20 dB, the contribution rate of the principal components based on Hilbert transform and the improved PCA algorithm is shown in Figure 6, in which the number of principal components is represented by the abscissa coordinate and the contribution rate to the corresponding principal component is represented by the vertical coordinate.

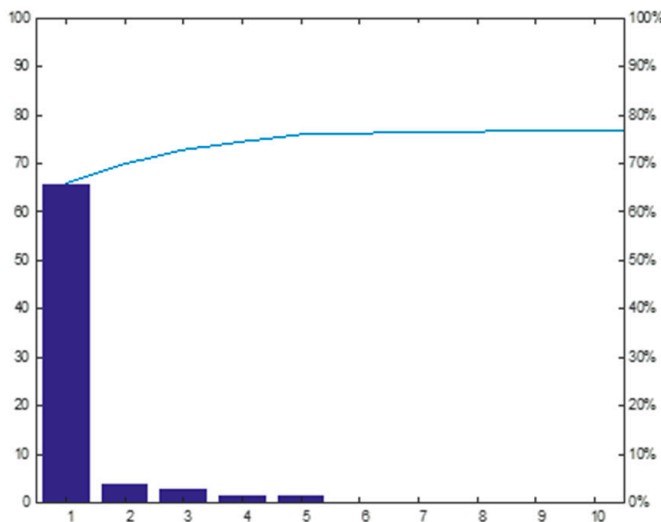

**Figure 6.** Histogram of the contribution rate of the principal components based on Hilbert transform and the improved PCA algorithm in the noisy environment with the SNR of 20 dB.

The experimental results show that the contribution rate of the first principal component based on Hilbert transform and traditional PCA algorithm is 60.869%, the contribution rate of the second principal component is 4.895%, the contribution rate of the third principal component is 3.019%, the contribution rate of the fourth principal component is 1.952%, and the cumulative contribution rate of the first four principal components is 70.735%. The contribution rate of the first principal component based on Hilbert transform and the improved PCA extraction algorithm is 65.893%, and the contribution rate of the second principal component is 4.054%, and the contribution rate of the third principal component is 2.912%, and the contribution rate of the fourth principal component is 1.656%, and the cumulative contribution rate of the first four principal components is 74.515%.

Based on the experimental results above, the performance of the traditional PCA algorithm and the improved PCA algorithm for radiation source individual classification and recognition is summarized. The contribution rates of the principal components of the traditional PCA algorithm and the improved PCA algorithm are shown in Table 1.

**Table 1.** The contribution rates of the principal components of the traditional PCA algorithm and the improved PCA algorithm.

| Feature Extraction Mode | Signal to Noise Ratio | First Principal Component | Second Principal Component | Third Principal Component | Fourth Principal Component | Cumulative Principal Components |
|---|---|---|---|---|---|---|
| Hilbert + Traditional PCA | 20 dB | 60.869% | 4.895% | 3.019% | 1.952% | 70.735% |
| Hilbert + Improved PCA | 20 dB | 65.893% | 4.054% | 2.912% | 1.656% | 74.515% |

From the experimental results in Table 1, it can be seen that the contribution rate of the principal components based on the improved PCA algorithm is obviously higher than that of the traditional PCA algorithm. That is to say, under certain conditions, the subtle features of the signals can be extracted better with less principal component feature dimension based on the improved PCA algorithm. Furthermore, the recognition effect is improved. The recognition results obtained when comparing two other traditional algorithms [25,26] in the environment with the SNR of 20 dB are shown in Table 2.

**Table 2.** Comparison of recognition rates in the environment of 20 dB.

| Recognition Mode | SNR | The Number of Principal Components Used | Recognition Rate |
|---|---|---|---|
| Hilbert + Traditional PCA | 20 dB | 2 | 94% |
| Hilbert + Improved PCA | 20 dB | 2 | 99.67% |
| Fractal Dimension Based Algorithm | 20 dB | 7 | 84.26% |
| Holder Theory Based Algorithm | 20 dB | 2 | 74.25% |

The experimental results obtained by calculating the recognition results under a different SNR environment with the improved PCA algorithm are reported in Table 3.

**Table 3.** Recognition results under different SNR environments calculated with the improved PCA algorithm.

| SNR | 20 dB | 18 dB | 16 dB | 14 dB |
|---|---|---|---|---|
| Recognition Results | 99.67% | 85.32% | 70.24% | 56.82 |

On the basis of the above data, it can be concluded that the improved PCA algorithm with meanization method can improve the contribution rate of the principal components and the recognition rate, which is beneficial to the identification of radiation source individual equipment, compared with the traditional PCA algorithm, fractal dimension-based algorithm, and Holder theory-based algorithm. The improved PCA algorithm has some advantages for the extraction of the subtle features of the radiation source individual signals. With the decrease of SNR, the recognition rate by the improved PCA algorithm also decreases quickly, because the fingerprint's features are very subtle and can easily to covered by the noise. Therefore, a future challenge is how to further extract the subtle features under lower SNR.

## 5. Conclusions

In order to solve the problem that is difficult to obtain and identify radiation source individual information accurately in complex electromagnetic environment, this paper designed and implemented a principal component analysis algorithm with meanization method, which was used to extract the subtle

features of radiation source individual signals from 10 identical Motorola walkie-talkies from the same manufacturer. Compared with the traditional PCA algorithm, this improved algorithm can improve the contribution rate of the principal components, reduce the dimension of extracting signal features, and improve the recognition rate, which is conducive to accurate identification of radiation source individual devices. Compared with the traditional algorithm, it has more advantages. But the PCA algorithm with meanization method is not applicable in some cases. For example, when the contribution rate of the principal component by the traditional PCA algorithm is high enough to represent most of the information of the original data, the improvement of the improved PCA algorithm will be reduced. To a certain extent, the improved PCA algorithm will increase the computational complexity, which is harmful to real-time performance and efficient identification of radiation source individual devices. However, it provides a powerful basis for the off-line application of signal recognition in related fields, such as electronic reconnaissance, image processing, fault diagnosis and so on.

**Author Contributions:** J.L. and D.B. participated in data analysis and the design of the study and drafted the manuscript; Y.Y. proposed the improved feature extraction algorithm; K.W. and B.Z. participate in the modification of the revised paper. All authors gave final approval for publication.

**Funding:** This research was supported by the National Natural Science Foundation of China (No. 51806135 and No. 61603239).

**Conflicts of Interest:** The authors declare no conflict of interest.

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
