# Peer review of "An Improved Algorithm for Extracting Subtle Features of Radiation Source Individual Signals"

_electronics, doi:10.3390/electronics8020246_

Round 1

Reviewer 1 Report

The work concerns the improvement of principal component analysis (PCA). The presented work is important from the point of view of pratical applications. Literature on this subject is rich. Therefore, in the introduction of the work, the state of literature should be supplemented according to the declared topic of work. There is a lack of comparison of the results of the analyzed PCA algorithm with other important ones in this topic. There is nothing new in subsection 3.1 line 74. What was the purpose of the description of the known Hilbert transform, since these expressions are not used later. There is a lack of linguistic references to analytical expressions, which makes it difficult to assess their originality. This should be completed. The editing of the work should be corrected, eg lines 162, 166, 169 contain the same numbering of markings as the analytical expressions placed in the work.

Author Response

Comments and Suggestions for Authors:

The work concerns the improvement of principal component analysis (PCA). The presented work is important from the point of view of pratical applications. Literature on this subject is rich. Therefore, in the introduction of the work, the state of literature should be supplemented according to the declared topic of work.

Answers: Thank you for your suggestions. We have already added more literatures in the paper.

There is a lack of comparison of the results of the analyzed PCA algorithm with other important ones in this topic. 

Answers: We already added two traditional feature extraction algorithms (Fractal dimension based algorithm and Holder theory based algorithm) to compared with the improved PCA algorithm, which shows in table 2.

There is nothing new in subsection 3.1 line 74. What was the purpose of the description of the known Hilbert transform, since these expressions are not used later.

Answers: Thank you for your advice. We deleted the part 3.1 already.

There is a lack of linguistic references to analytical expressions, which makes it difficult to assess their originality. This should be completed.

Answers: Thank you for your suggestions. We already added some in the manuscirpt.

The editing of the work should be corrected, eg lines 162, 166, 169 contain the same numbering of markings as the analytical expressions placed in the work.

Answers: We correct the formulas in lines 162,166,169 in the manuscript and checked all the formulas in the whole manuscript.

Reviewer 2 Report

a 20dB SNR has been introduced as a particularly harsh environment. What about further decreasing such value ? what is the limit for any classification ? what is the value that you retain as still acceptable ?

In other words, a parametric study is necessary ....

Further, can the authors provide a real test bed with actual signals transmitted and received, instead of using 'synthetic' environments ?

Author Response

Comments and Suggestions for Authors:

a 20dB SNR has been introduced as a particularly harsh environment. What about further decreasing such value ? what is the limit for any classification ? what is the value that you retain as still acceptable ?

in other words, a parametric study is necessary ....

Answers: Thank you for your suggestions. We calculated the recognition results under the SNR of 20dB, 18dB, 16dB, 14dB, and when the SNR decreasing, the recognition results decreasing quickly, because the fingerprint features of the signals are too subtle to extract, we have already analyzed the reasons in the manuscript.

Further, can the authors provide a real test bed with actual signals transmitted and received, instead of using 'synthetic' environments ?

Answers: In fact, the experimental part of the paper uses a real experimental test environment. The data tested is also the result of real measurements, and there is no synthetic environment. In the experimental part of the article, we have emphasized the experimental conditions as follows:

10 sets of transient signals of radio equipment are collected. Data was collected with Agilent oscilloscope and was injected directly without antenna.Each station equipment took 50 groups of data, sampling rate is 40 MHz, each group of data collect 159,901 points.

Round 2

Reviewer 2 Report

no further comments